# Impact of the First Wave of the COVID-19 Pandemic on the Lyon University Hospital Cancer Institute (IC-HCL)

**DOI:** 10.3390/cancers14010029

**Published:** 2021-12-22

**Authors:** Anne-Sophie Belmont, Christophe Sajous, Amandine Bruyas, Sara Calattini, Stéphanie Cartalat, Marion Chauvenet, Marc Colombel, Stéphane Dalle, Tristan Dagonneau, Marie Darrason, Gilles Devouassoux, Michaël Duruisseaux, Marielle Guillet, Olivier Glehen, Pierre Philouze, François Tronc, Thomas Walter, Benoît You, Gilles Freyer

**Affiliations:** 1Plateforme Transversale de Recherche Clinique de l’Institut de Cancérologie, Hospices Civils de Lyon, 69310 Pierre-Benite, France; sara.calattini@chu-lyon.fr; 2Service d’Oncologie Médicale, Hôpital Lyon Sud, Hospices Civils de Lyon, 69310 Pierre-Benite, France; christophe.sajous@chu-lyon.fr (C.S.); benoit.you@chu-lyon.fr (B.Y.); 3Service d’Oncologie Médicale, Hôpital de la Croix-Rousse, Hospices Civils de Lyon, 69004 Lyon, France; amandine.bruyas@chu-lyon.fr; 4Service de Neuro-Oncologie, Hôpital Pierre Wertheimer, Hospices Civils de Lyon, 69500 Bron, France; stephanie.cartalat@chu-lyon.fr; 5Service d’Hépato-Gastroentérologie, Hôpital Lyon Sud, Hospices Civils de Lyon, 69310 Pierre-Benite, France; marion.chauvenet@chu-lyon.fr; 6Service d’Urologie et Chirurgie de la Transplantation, Hôpital Edouard Herriot, Hospices Civils de Lyon, 69003 Lyon, France; marc.colombel@chu-lyon.fr; 7Service de Dermatologie, Hôpital Lyon Sud, Hospices Civils de Lyon, 69310 Pierre-Benite, France; stephane.dalle@chu-lyon.fr; 8Départment d’Information Médicale, Hôpital Lyon Sud, Hospices Civils de Lyon, 69310 Pierre-Benite, France; tristan.dagonneau@chu-lyon.fr; 9Service de Pneumologie Aiguë Spécialisée et Cancérologie Thoracique, Hôpital Lyon Sud, Hospices Civils de Lyon, 69310 Pierre-Benite, France; marie.darrason@chu-lyon.fr; 10Service de Pneumologie, Hôpital de la Croix-Rousse, Hospices Civils de Lyon, 69004 Lyon, France; gilles.devouassoux@chu-lyon.fr; 11Service de Pneumologie, Hôpital Louis Pradel, Hospices Civils de Lyon, 69500 Bron, France; michael.duruisseaux@chu-lyon.fr; 12Service d’Hépato-Gastroentérologie, Hôpital de la Croix-Rousse, Hospices Civils de Lyon, 69004 Lyon, France; marielle.guillet@chu-lyon.fr; 13Service de Chirurgie Digestive et Oncologique, Hôpital Lyon Sud, Hospices Civils de Lyon, 69310 Pierre-Benite, France; olivier.glehen@chu-lyon.fr; 14Service d’Oto-Rhino-Laryngologie et Chirurgie Cervico-Faciale, Hôpital de la Croix-Rousse, Hospices Civils de Lyon, 69004 Lyon, France; pierre.philouze@chu-lyon.fr; 15Service Chirurgie Thoracique Vidéothoracoscopie et Transplantation Pulmonaire, Hôpital Louis Pradel, Hospices Civils de Lyon, 69500 Bron, France; francois.tronc@chu-lyon.fr; 16Service d’Oncologie Médicale, Hôpital Edouard Herriot, Hospices Civils de Lyon, 69003 Lyon, France; thomas.walter@chu-lyon.fr

**Keywords:** COVID-19, solid cancers, hospital activity, protective measures, treatment adjustments, treatment dispensations, MDTMs, surgery activities, clinical trials, mortality

## Abstract

**Simple Summary:**

This article presents the protective measures put in place at the “Institut de Cancérologie des Hospices de Lyon” (IC-HCL) during the first wave of the COVID-19 pandemic in France (spring 2020) and how they impacted IC-HCL clinical activity. Spring 2020 activities were compared to winter 2019–2020.

**Abstract:**

This article presents the protective measures put in place at the “Institut de Cancérologie des Hospices de Lyon” (IC-HCL) during the first wave of the COVID-19 pandemic in France (spring 2020) and how they impacted IC-HCL clinical activity. Spring 2020 activities were compared to winter 2019–2020. Results showed a decrease of activity of 9% for treatment dispensations, 17% for multidisciplinary team meetings, 20% for head and neck and thoracic surgeries, and 58% for new patient enrolment in clinical trials. Characteristics of patients treated for solid cancer and hospitalized for COVID-19 during spring 2020 were collected in a retrospective study. Mortality was attributed to COVID-19 for half of the cases, 82% being patients above 70 and 73% being stage IV. This is in concordance with current findings concluding that the risk of developing severe or critical symptoms of COVID-19 is correlated with factors co-occurring in cancer patients and not to the cancer condition per se. While a number of routines and treatment regimens were changed, there was no major decline in numbers of treatments conducted at the IC-HCL during the first wave of the COVID-19 pandemic that hit France between March and May 2020, except for clinical trials and some surgery activities.

## 1. Introduction

On 31 December 2019, the World Health Organization (WHO) was informed about the first “cases of pneumonia of unknown etiology” (later identified as the SARS-CoV-2 (COVID-19)). The virus was confirmed to have reached Europe on 24 January 2020, when the first COVID-19 case was identified in France. On 30 January 2020, the virus was identified in Italy, followed by a wave of massive infections in the north of Italy, putting its healthcare system on the edge of breaking. Two weeks later, a similar situation was observed in France, with an unprecedented swell of patients’ hospitalization, ICU beds saturation, and medical staff shortage, leading to a total lockdown between 17 March 2020 and 11 May 2020.

The way in which to manage cancer patients and pursue oncological practice in this context was of high challenge. In China, due to the complexity of the situation, it was decided to postpone postoperative chemotherapy, to withhold chemotherapy when concomitantly prescribed with radiation, and to reduce chemotherapy doses in older and/or frail patients. In France, the first guidelines to ensure patient protection and make oncology departments “sanctuaries” free of COVID-19 were released on 25 March 2020 [1]. The recommendations were largely based on the Chinese experience, but also on worrisome data from Italy.

Here, we present how these guidelines were implemented at the “Institut de Cancérologie des Hospices de Lyon” (IC-HCL), a French hospital overload with COVID-19 patients during the peak of the pandemic (March–May 2020). The measures put in place to protect patients and caregivers are described in a narrative summary as well as how these measures have impacted the IC-HCL activities in terms of treatment sessions, consultations, surgical procedures, and multidisciplinary team meetings (MDTMs). The number of patients with solid cancer hospitalized for COVID-19 during this period was also collected as their characteristics in terms of cancer type, stage, and co-morbidities.

## 2. Materials and Methods

Period of reference: Spring 2020 corresponds to the 3 month period of the first wave of the COVID-19 pandemic in France: March 2020 to May 2020. Winter 2019–2020 corresponds to the 3 month period from December 2019 to February 2020. Spring 2019 corresponds to the 3 month period from March 2019 to May 2019. Data were collected for spring 2020, winter 2019–2020, and spring 2019. Here, we defined winter 2019-2020 (the three months immediately before COVID-19 pandemic) as our period of reference, compared to spring 2020. Comparison between spring 2020 and spring 2019 activities are provided in some cases for complementary information.

Data were collected for solid tumor cancer patients only.

Strategies put in place to avoid cross-contamination between patients and caregivers, and treatment adjustments: Ten heads of oncology departments from different specialties (pulmonology, digestive, medical oncology and gynecology, dermatology, neuro-oncology) were asked to fill an electronic questionnaire concerning the measures taken in spring 2020 to counter the COVID-19 epidemic within the hospital. 

Treatment dispensations: Data were collected from the 4 oncological treatment platforms of the IC-HCL. Collected data included IV systemic therapies (chemotherapies, immunotherapies, etc.). Radiotherapies were not included in this dataset (impossibility for radiotherapists to provide these data for the concerned periods).

MDTMs: The number of patient files discussed per month were collected from the 17 MDTMs (10 specialties) that are run weekly at the IC-HCL.

Consultations: The number of physical and teleconsultations (phone) per week were provided by the medical information department. Aside from the medical oncology department, in the other departments, it was not possible to distinguish oncological from non-oncological consultations.

Surgical procedures: Data were collected for the head and neck and thoracic surgery departments. For gynecology procedures, data were extrapolated from Lamblin et al. [2]. Data from other specialties were not available for technical reasons (impossibility to distinguish cancer-related surgeries from others). The number of HCL operating rooms that remained opened during spring 2020 was extracted from an article by Philouze et al. [3].

Clinical trials: The status of HCL-sponsored trials during spring 2020 (opened or closed) was provided by the HCL research department. Monthly trial enrolment rate for cancer trials (HCL-sponsored trials and external sponsored trials) were provided by the IC-HCL clinical research assistants from each specialty. 

Patients: The number of patients hospitalized for COVID-19 during spring 2020 was provided by the medical information department. Among them, a retrospective study (NCT04910633) was conducted on patients with an “active” solid cancer (i.e., patients who were following an oncological treatment). The following data were collected: sex, age, cancer type, cancer stage, type of anti-cancer treatment, nature of comorbidities (3 maximum), vital status, COVID-19 severity (based on chest X-ray damage (low ≤ 25%, medium: between 25 and 50%, high > 50%) and for those who died, causality between COVID-19 status and death.

## 3. Results

### 3.1. Strategies Put in Place to Avoid Cross-Contamination between Patients and Caregivers

In early March 2020, the following measures were put in place in oncology departments: phone calls prior to patient hospital visit, in order to verify the absence of COVID-19 symptoms; request for patient to come alone; assignment of dedicated caregivers to screen COVID-19 symptoms using questionnaires and measuring patient’s temperature; reorganization of waiting rooms to respect social distancing; mandatory hand sanitization and protective face masks for both patients and professionals; installation of curtains and screens to avoid cross-contamination between roommates for double rooms; set-up of a dedicated space organized for patients with severe COVID-19 symptoms (low blood oxygen, important coughing, difficulty breathing, etc.) prior to hospitalization; and maintaining of doors being closed for patients that were requested to not circulate outside their room.

### 3.2. Treatment Adjustments

The reported treatment adjustments made at the IC-HCL were as follows: switch from IV to oral therapy whenever possible (5FU was replaced by capecitabine; prescription of temporary hormone therapy for HER-positive breast cancer patients); modification of the administration schedule to reduce the frequency of treatment administration (pembrolizumab 200 mg every three weeks was replaced by pembrolizumab 400 mg every 6 weeks; nivolumab 240 mg every two weeks was replaced by nivolumab 480mg every month; weekly 80 mg/m^2^ paclitaxel was replaced by 175 mg/m^2^ paclitaxel with granulocyte colony-stimulating factor (G-CSF) every 3 weeks; cancelation of 1/2 trastuzumab/cetuximab administration for patients having received maintenance treatment for several months/year(s)); switch of the type of the molecule to reduce the duration of the hospitalization (cisplatin was replaced by carboplatin whenever possible); and generalization of G-CSF prescription—even in patients with low-risk of febrile neutropenia.

### 3.3. Treatment Dispensations

A total of 6960 oncological treatment sessions were conducted during spring 2019—they were 7400 during winter 2019–2020 and 6699 during spring 2020. The decrease of activity between spring 2020 and the period of reference (winter 2019–2020) was 9% (Figure 1). 

The most impacted specialties were head and neck (−19%), digestive (−14%), and dermatology (−14%). Medical oncology and thoracic activities were respectively impacted by −8%, and −5%. Neurology was not impacted (0%; Figure 2).

### 3.4. Multidisciplinary Team Meetings (MDTMs)

In total, there was a decrease of 17% of MDTM activity. The most impacted specialties were dermatology (31%), urology (31%), thyroid (28%), gynecology (27%), and medical oncology (24%), followed by digestive (18%), neurology (18%), thoracic (15%), and endocrine tumors (12%). The least impacted specialty was head and neck (7%; Table 1).

### 3.5. Consultations

Teleconsultations represented 0% of all consultations in January (0/1221), 0% in February 2020 (0/1148), 29% in March (407/1381), 42% in April (430/1035), and 24% in May (217/919).

### 3.6. Surgical Procedures for Cancer Treatment

A total of 173 head and neck surgeries were conducted during spring 2019—they were 203 during winter 2019–2020 and 158 during spring 2020. The decrease of activity between spring 2020 and the period of reference (winter 2019–2020) was 22%.

For thoracic surgeries, the decrease of activity was 18% (Table 2). 

Criteria to postpone surgery were decided by each department depending on operating room and staff availability of each IC-HCL site and the patient’s specific condition.

During spring 2020, 3 surgeries were postponed/cancelled for head and neck cancers (161 planned versus 158 performed) and 19 for thoracic cancers (209 planned versus 190 performed). Moreover, among the 115 gynecological surgical procedures planned, 40 (34.8%) were postponed, 7 (6.1%) were cancelled, and 9 (7.8%) had been modified in comparison with the initial surgery strategy [2]. In total, the number of operating rooms was reduced from 20 to 5 as they were gradually reassigned to new COVID-19 intensive care units [3].

### 3.7. Clinical Trials

They were 244 HCL-sponsored clinical trials opened to inclusion during winter 2019–2020 (29/224 being dedicated to cancer patients). In total, 224/244 (91%) were suspended at the beginning of March 2020. Among the 20 trials that remained opened, 11 were cancer trials.

During spring 2020, 39 patients were included in cancer trials (HCL-sponsored trial and external sponsored trials) compared to 92 during winter 2019–2020 and 93 during spring 2019. The inclusion rate decrease was 58% compared to the period of reference (Figure 3).

### 3.8. Patients

A total of 1781 patients were hospitalized at the HCL for COVID-19 during spring 2020; 333 died (18.7%). Our retrospective study identified 43 patients that were following an oncological treatment for solid cancer during this period; 51% died (22/43). Half of the deaths were linked to COVID-19 (11/22), and the other half to cancer progression (11/22). For the deaths linked to COVID-19, 82% of patients were above 70 years old (9/11) and 73% were stage IV (8/11; Figure 4). Lung cancer was the most represented cancer type, with 36% of patients (4/11; Figure 4). All patients were suffering from at least one comorbidity, with hypertension being the most representative (73%; 8/11), followed by diabetes (45%; 5/11) and vascular disorders (36%; 4/11).

## 4. Discussion

When the COVID-19 pandemic outbreak reached Europe, oncological learned societies quickly released guidelines to ensure patient protection [1,4,5,6,7,8,9,10,11,12,13,14,15,16,17,18]. Recommendations were made to minimize patient presence at the hospital; it was recommended to prefer online consultations, to use oral instead of intravenous drugs, to interspace drugs and irradiation scheduling, and to allow temporary breaks in the treatment of advanced-line patients with chronic diseases. The authors also suggested to prioritize hospital visits based on curative intent, age, life-expectancy, earlier line of treatment, severity of symptoms, and palliative care need. The results of the present study indicate that the adjustments made at the IC-HCL are totally in line with these recommendations. These have, however, not greatly impacted the activity of the IC-HCL in terms of the number treatment dispensations. 

It would be of interest to compare the effect of the approach used in the IC-HCL to that of other centers in Europe, but, to the best of our knowledge, similar data have yet to be published. 

IV treatment administrations schedules were not greatly modified, and patients continued to come to the hospital to receive their treatment. However, medical consultations were mostly switched to teleconsultations, especially during the highest peak of the pandemic in April 2020. This avoided changes in the therapeutic plan of patients that is reported to lead to high level of anxiety, distress, and fear [19]. Unlike IV treatments, cancer surgery was differently impacted by COVID-19 depending on specialty. Indeed, thoracic and head and neck operations were not greatly impacted by the COVID-19 epidemic, while half of the gynecological surgeries were cancelled because of the COVID-19 outbreak [2]. This surgical triage was perhaps not optimal as it is now proven that a 12 week surgery delay results in overall survival being decreased for breast cancer patients [20].

Accordingly, the number of patients’ files discussed at the MDTMs did not decrease significantly for head and neck and thoracic specialties compared to gynecological specialties. Data concerning the other specialties are needed in order to confirm this result. 

It is of note that clinical research was the most impacted activity in our institution, as almost all HCL-sponsored trials were suspended. This is concordant with the report by Upadhaya et al. who found that only 20% of the research institutions in the United States and 14% of the research institutions in Europe continued to enroll patients at their usual rate during the first wave of the COVID-19 pandemic period [21].

Regarding the link between COVID-19 and the mortality of cancer patients, the results of the present small study are in concordance with the current understanding that adverse COVID-19 outcomes are principally driven by advanced stage, age, and the presence of other non-cancer comorbidities and that the risk to develop severe or critical symptoms of COVID-19 is correlated with factors co-occurring in cancer patients and not to the cancer condition per se [22,23,24,25,26]. The results of the present study also correlate with the findings that lung cancer seems to be an important risk factor for COVID-19 severity, certainly due to factors related to underlying pulmonary condition, as well as lower acceptance rate in critical care [27,28]. Our study has important limitations (small size sample, observational results only, no control cases, first wave patients not affected by new variants) that, however, restrict these conclusions.

## 5. Conclusions

While a number of routines and treatment regimens were changed, there was no major decline in numbers of treatments conducted at the IC-HCL during the first wave of the COVID-19 pandemic that hit France between March and May 2020, except for clinical trials and some surgery acts. For the next waves, it would be of interest to maintain the research activity to prevent drug discovery attrition and patients access to new treatments. The modification of surgical triage guidelines for future COVID-19 waves should also be implemented for certain specialties to limit patients’ loss of chance.

## Figures and Tables

**Figure 1 cancers-14-00029-f001:**
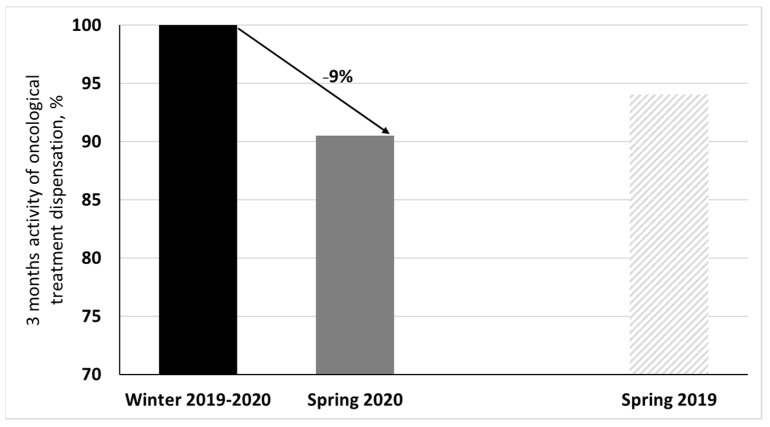
Number of oncological treatment sessions realized in spring 2020 (and spring 2019) compared to winter 2019–2020 as 100% reference.

**Figure 2 cancers-14-00029-f002:**
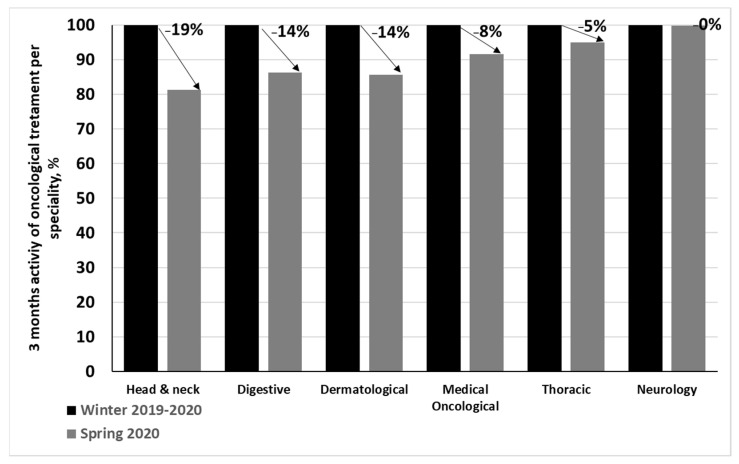
Number of oncological treatment sessions realized per specialty in spring 2020 compared to winter 2019–2020 as 100% reference.

**Figure 3 cancers-14-00029-f003:**
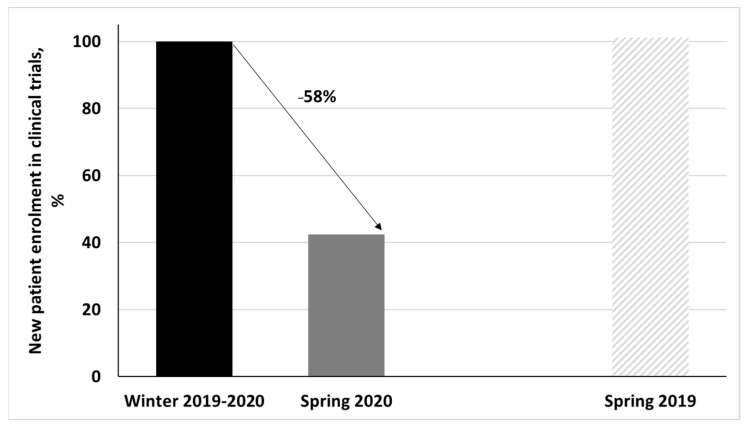
Number of new patients enrolled in cancer clinical trials in spring 2020 (and spring 2019) compared to winter 2019–2020 as 100% reference.

**Figure 4 cancers-14-00029-f004:**
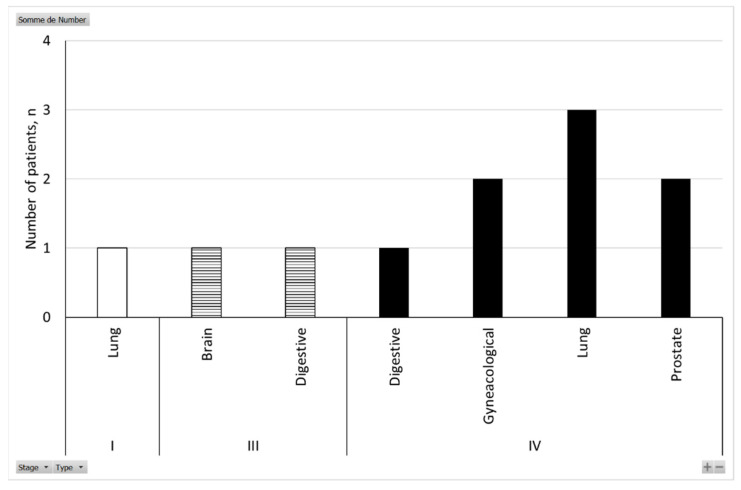
Cancer stage and type of patients who died from COVID-19.

**Table 1 cancers-14-00029-t001:** Number of patients files discussed per month during the 10 IC-HCL MDTMs. Spring 2020 data are compared to winter 2019–2020 as 100% reference.

Specialty	Winter 2019–2020				Spring 2020				% of Activity Decrease
	19 Dec	20 Jan	20 Feb	Total	20 March	20 April	20 May	Total	
Dermatology; n	117	157	127	401	102	63	112	277	31
Urology; n	129	196	211	536	126	143	101	370	31
Thyroid; n	61	60	73	194	73	38	29	140	28
Gynecology; n	193	176	158	527	158	119	106	383	27
Medical oncology; n	99	123	110	332	79	88	84	251	24
Digestive; n	260	370	305	935	292	261	211	764	18
Neurology; n	94	144	105	343	100	89	92	281	18
Thoracic; n	237	259	260	756	263	188	193	644	15
Endocrine; n	62	71	64	197	72	60	42	174	12
Head and neck; n	31	30	38	99	38	29	25	92	7
TOTAL	1283	1526	1451	2662	1303	1040	995	2206	17

**Table 2 cancers-14-00029-t002:** Number of surgical procedures realized over spring 2019, winter 2019–2020, and spring 2020.

Specialty	Number of Surgeries, Spring 2019	Number of Surgeries, Winter 2019–2020	Number of Surgeries, Spring 2020	Activity Decrease (%) (Spring 2020 Compared to Winter 2019–2020)	Activity Decrease (%) between Spring 2020 and Spring 2019
Head and neck, n	173	203	158	22	9
Thoracic, n	201	232	190	18	5

## Data Availability

The data presented in this study are available on request from the corresponding authors. The data are not publicly available due to the necessity to make every effort to maintain patients non identifiable according to European law.

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
