# Peer review of "Impact of the First Wave of the COVID-19 Pandemic on the Lyon University Hospital Cancer Institute (IC-HCL)"

_cancers, 2021, doi:10.3390/cancers14010029_

Round 1

Reviewer 1 Report

Dear authors,

You provide some well-researched data on the changes implemented in your hospital during the first wave of the COVID pandemic.

While I generally believe that you present some interesting data, I have the following suggestions:

- If you want to provide some “best practice” patterns, I would suggest to be more specific with regards to the measures implemented: What online tools did you use for consultations? How was the determination made to postpone surgeries (via the surgical department or via MDTMs), did you have pre-specified criteria for this?

- You changed chemotherapy treatment as well as surgery. Was radiation treatment – which typically includes a substantial number of outpatient visits – also impacted?

- “Treatment dispensation”: What does that term exactly refer to: Applications of systemic therapy only or did it also include consultations without therapy application? This seems to be an interesting point given the wide-ranging changes in treatment patterns described in 3.2. and the comparably small impact (-9%). It would be great to have a comment on this.

- Based on the comparisons, it seems that the hospital grew between Spring 2019 and Winter 2020 (Fig.1, Table 2). Is that the reason why treatments and surgeries are more similar between spring 2019 and spring 2020 or do you generally have more visits and surgeries in winter? A small comment to understand differences between spring 2019 and winter 2020 would be helpful.

- In your methods, you note that you collected COVID-19 severity (based on chest X-rays). However, in the results section, I do not find any analyses for this characteristic. What did you find regarding COVID-19 severity in the 43 cancer patients?

- I would suggest rephrasing you conclusion that “activities were not greatly impacted” to reflect that, while a number of routines and treatment regimens were changed, there was no major decline in numbers of treatment (meaning COVID had an impact, just not a numerical one).

Minor points:

- If I understand Figure 4 correctly, 5 of 11 cancer patients with COVID-related deaths had lung cancer, as opposed to four, as you suggest (l. 224). Please clarify.

- It seems that 430/1035 does not match the 71% figure in line 185.

- There may be a word missing in line 167.

Author Response

Please see Cover letter attached

Reviewer 2 Report

The manuscript entitled “COVID & CANCER: impact of the first wave of the COVID-19 pandemic on the Lyon University Hospital Cancer Institute (LUH-CI)” presents the protective measures put in place at the LUH during the first wave of the COVID-19 pandemic in France (spring 2020) and how they impacted LUH-CI clinical activity.

General Critique:

From a general standpoint, the current study is lack of the standard clinical research criteria, including study strategy, and how to assess or validate the impaction of in a cancer hospital? So, in current form of manuscript is like a summarized narrative hospital management report during first wave COVID-19 pandemic. However, this is a relatively well written study, but does have some shortcomings which limit its conclusions and potential for publication in a high impact journal such as cancers.

The basic question is interesting. However, I would like to raise the following points:

  1. The section of Introduction is too simplified, not well-organized, and not focus the impaction of clinical practice during COVID-19 pandemic in the literature. The author should be stated more clearly, thoroughly, and clearly state research hypothesis of the study.
  1. Brief the redundant description in “Materials and methods” section and integrate the results into a compact format. Please perform statistical analysis to interpret research results.
  2. The manuscript should be carefully revised in regard to grammatical errors, and the qualities of the figures, table and the whole manuscript need to be improved.

Author Response

Please see cover letter attached

Round 2

Reviewer 1 Report

The manuscript is much improved and I recommend publication.

Reviewer 2 Report

The manuscript entitled “COVID & CANCER: impact of the first wave of the 2 COVID-19 pandemic on the Lyon University Hospital 3 Cancer Institute” presents the protective measures put in place at the LUH during the first wave of the COVID-19 pandemic in France (spring 2020) and how they impacted LUH-CI clinical activity. The editor decided to second review on the revised manuscript

General Critique:

From a general standpoint, the revised manuscript is much improved and well written, but does have some shortcomings which limit its conclusions and potential for publication in a high impact journal such as cancers. The current study is like a summarized narrative hospital management report during first wave COVID-19 pandemic. The basic question is interesting. However, I would not like to endorse for publication.